# COVID-19 Salivary Protein Profile: Unravelling Molecular Aspects of SARS-CoV-2 Infection

**DOI:** 10.3390/jcm11195571

**Published:** 2022-09-22

**Authors:** Eduardo Esteves, Vera M. Mendes, Bruno Manadas, Rafaela Lopes, Liliana Bernardino, Maria José Correia, Marlene Barros, Ana Cristina Esteves, Nuno Rosa

**Affiliations:** 1Universidade Católica Portuguesa, Faculty of Dental Medicine (FMD), Center for Interdisciplinary Research in Health (CIIS), 3504-505 Viseu, Portugal; 2Health Sciences Research Centre (CICS-UBI), Faculty of Health Sciences, University of Beira Interior, 6201-001 Covilhã, Portugal; 3CNC—Center for Neuroscience and Cell Biology, CIBB—Centre for Innovative Biomedicine and Biotechnology, University of Coimbra, 3004-504 Coimbra, Portugal; 4CESAM, Department of Biology, University of Aveiro, 3810-193 Aveiro, Portugal

**Keywords:** COVID-19, saliva, proteomics, interactomics, Oralint

## Abstract

COVID-19 is the most impacting global pandemic of all time, with over 600 million infected and 6.5 million deaths worldwide, in addition to an unprecedented economic impact. Despite the many advances in scientific knowledge about the disease, much remains to be clarified about the molecular alterations induced by SARS-CoV-2 infection. In this work, we present a hybrid proteomics and in silico interactomics strategy to establish a COVID-19 salivary protein profile. Data are available via ProteomeXchange with identifier PXD036571. The differential proteome was narrowed down by the Partial Least-Squares Discriminant Analysis and enrichment analysis was performed with FunRich. In parallel, OralInt was used to determine interspecies Protein-Protein Interactions between humans and SARS-CoV-2. Five dysregulated biological processes were identified in the COVID-19 proteome profile: Apoptosis, Energy Pathways, Immune Response, Protein Metabolism and Transport. We identified 10 proteins (KLK 11, IMPA2, ANXA7, PLP2, IGLV2-11, IGHV3-43D, IGKV2-24, TMEM165, VSIG10 and PHB2) that had never been associated with SARS-CoV-2 infection, representing new evidence of the impact of COVID-19. Interactomics analysis showed viral influence on the host immune response, mainly through interaction with the degranulation of neutrophils. The virus alters the host’s energy metabolism and interferes with apoptosis mechanisms.

## 1. Introduction

The emergence of the novel coronavirus disease 2019 (COVID-19), caused by the infection with SARS-CoV-2 coronavirus, created the urgency to develop new diagnostic and therapeutic targets, as well as identify preventive strategies [1]. PCR-based diagnostics can be implemented and scaled quickly, but do not provide information on disease mechanisms [2,3].

Mass spectrometry (MS)-based proteomics does not depend on previous information and can be set up in an untargeted way. The sensitive and high-throughput mass spectrometers allow detection of even the faintest changes in host physiology. Currently, MS-based proteomic workflows are well established, and can be routinely used for definition of molecular pathways and pathogenesis, prognostic’s biomarkers, drug discovery or repurposing and vaccine development [4,5]. Concerning COVID-19, MS-based proteomics has the potential to unveil aspects of the disease, providing valuable information for understanding the molecular pathways disrupted by virus infection [4]. 

Serum (and plasma) are, by a long shot, the most studied and well-known fluids with regards to human diseases. This over-burden of information has the inborn drawback of decreasing research in other biological fluids. Alternative fluids of diagnosis, such as saliva, have clear advantages when compared to blood [6,7]. Saliva has grown in popularity as a fluid of interest in recent years, due to its ease of collection via a non-invasive sampling procedure. Overall, saliva can be used to shed insight on illness development and pathophysiological mechanisms, in several (oral and systemic) diseases, including SARS-CoV-2 infection [8,9,10,11,12].

While proteomics has already proven to be a valuable tool for identifying biomarkers for many conditions, more strategies are still needed to functionally interpret proteomics data to clarify molecular changes of diseases such as COVID-19 and predict disease severity [11,13].

We hypothesize that SARS-CoV-2 induces characteristic proteome changes that can be detected in the saliva of infected patients. These molecular changes may shed light on the association of infection to host response.

## 2. Materials and Methods

### 2.1. Ethical Statement

This study was carried out in accordance with the Helsinki Declaration and the Oviedo Convention. The ethical aspects of this study were reviewed and approved by the Ethics Committee for Health at Centro Hospitalar Tondela Viseu. Written informed consent including the purpose of the study, data confidentiality, rights of participants, and the right to withdraw from the study at any time was provided by every participant or by their legal representatives or guardians before study enrolment.

### 2.2. Participant Enrolment

Five saliva samples (2 females and 3 males, aged 16 to 71) of a cohort of COVID-19 patients at Centro Hospitalar Tondela Viseu were collected. Furthermore, saliva samples from five healthy subjects (2 females and 3 males, aged 28 to 59) showing no evidence of oral and systemic pathologies or inflammatory processes, negative for SARS-CoV-2, were also collected. All samples were tested by PCR for SARS-CoV-2 presence using a previously established protocol for SARS-CoV-2 testing in saliva [9].

The COVID-19 patients exhibited a variety of symptoms: acute respiratory insufficiency syndrome (n = 1), cough (n = 2), dyspnea (n = 1), headache (n = 2) and myalgia (n = 3).

### 2.3. Saliva Collection

Passive drooled saliva samples (2 mL) were collected in 50 mL sterile tubes without stabilizers as described [9] and in agreement with the Portuguese healthcare guidelines (DGS Norma 004 2020). Samples were refrigerated and inactivated using 1% Triton X-100 detergent. Whole saliva was centrifuged at 10,000× *g*, 10 min, 4 °C. The supernatant was aliquoted and stored at −80 °C for analysis.

### 2.4. Sample Preparation

Protein concentration was determined using the Pierce BCA assay kit (ThermoFisher Scientific, Waltham, MA, USA), according to the manufacturers’ instructions, and concentration was adjusted to 50 μg/mL. Saliva was analyzed by capillary electrophoresis (Experion™ automated capillary electrophoresis system, Bio-Rad, Hercules, CA, USA), and compared to the in-house profiles to confirm sample integrity. 

For data-dependent acquisition (DDA) experiments, replicates from each condition were pooled into two different samples (COVID+ and COVID-) before sample processing. For data-independent acquisition (DIA), each sample was processed individually. Protein content (60 µg) from each sample was separated by SDS-PAGE (4–15%) for about 17 minutes at 110 V (Short-GeLC Approach [14] and stained with Coomassie Brilliant Blue G-250. For DDA experiments, each lane was divided into 5 gel pieces, and for DIA experiments into 3 gel pieces for further individual processing. After the distaining step, gel bands were incubated overnight with trypsin for protein digestion and peptides were extracted from the gel using 3 solutions containing different percentages of acetonitrile (30, 50, and 98%) with 1% formic acid. Acetonitrile was evaporated using a vacuum-concentrator, and peptides were re-suspended in 25 µL (DDA) or 30 µL (DIA) 2% acetonitrile/0.1% formic acid. Each sample was sonicated using a cup-horn (Ultrasonic processor, 750 W) for about 2 min, 40% amplitude, and pulses of 1 s ON/OFF. Ten microliters of each sample (DDA) or 5 µL (DIA) were analyzed by LC-MS/MS.

### 2.5. LC-MS Methodology

Samples were analyzed on a NanoLC™ 425 System (Eksigent^®^) coupled to a Triple TOF™ 6600 mass spectrometer (Sciex^®^, USA) equipped with an ESI DuoSpray™ Source (Sciex^®^, USA). The chromatographic separation was performed on a Triart C18 Capillary Column 1/32″ (12 nm, S-3µm, 150 × 0.3 mm, YMC) and using a Triart C18 Capillary Guard Column (0.5 × 5 mm, 3 μm, 12 nm, YMC) at 50 °C. The flow rate was set to 5 µL/min and mobile phases A and B were 5% DMSO plus 0.1% formic acid in water and 5% DMSO plus 0.1% formic acid in acetonitrile, respectively. The LC program was performed as followed: 5–30% of B (0–50 min), 30–98% of B (50–52 min), 98% of B (52–54 min), 98–5% of B (54–56 min), and 5% of B (56–65 min). The ionization source was operated in the positive mode set to an ion spray voltage of 5500 V, 25 psi for nebulizer gas 1 (GS1), 10 psi for nebulizer gas 2 (GS2), 25 psi for the curtain gas (CUR), and source temperature (TEM) at 100 °C. For DDA experiments, the mass spectrometer was set to scanning full spectra (*m*/*z* 350–2250) for 250 ms, followed by up to 100 MS/MS scans (*m*/*z* 100–1500). Candidate ions with a charge state between +1 and +5 and counts above a minimum threshold of 10 counts per second were isolated for fragmentation, and one MS/MS spectrum was collected before adding those ions to the exclusion list for 15 seconds (mass spectrometer operated by Analyst^®^ TF 1.8.1, Sciex^®^). The rolling collision was used with a collision energy spread of 5. For SWATH experiments, the mass spectrometer was operated in a looped product ion mode and specifically tuned to a set of 42 overlapping windows, covering the precursor mass range of 350–1400 *m*/*z*. A 50 ms survey scan (350–2250 *m*/*z*) was acquired at the beginning of each cycle, and SWATH-MS/MS spectra were collected from 100–2250 *m*/*z* for 50 ms, resulting in a cycle time of 2.1 s.

### 2.6. Data Analysis

#### 2.6.1. Ion-Library Construction (DDA Information)

A specific ion-library of the precursor masses and fragment ions was created by combining all files from the DDA experiments in one protein identification search using the ProteinPilot™ software (v5.0, Sciex^®^, USA). The paragon method parameters were the following: searched against the reviewed Human (SwissProt) database, cysteine alkylation by acrylamide, digestion by trypsin, and gel-based ID. An independent false discovery rate (FDR) analysis, using the target-decoy approach provided by Protein Pilot™, was used to assess the quality of identifications.

#### 2.6.2. Relative Quantification of Proteins (SWATH-MS)

SWATH data processing was performed using SWATH^TM^ processing plug-in for PeakView^TM^ (v2.0.01, Sciex^®^). Protein relative quantification was performed in all samples using the information from the protein identification search. Quantification results were obtained for peptides with less than 1% of FDR and by the sum of up to 5 fragments/peptide. Each protein was normalized for the total sum of areas for the respective sample. Protein quantities were obtained by the sum of up to 15 peptides/protein (Appendix A).

### 2.7. Protein Functional Analysis

Functional analysis of altered salivary proteins in COVID-19 identified by mass spectrometry was performed according to Figure 1. 

Class enrichment (biological process) of COVID+ saliva proteins was obtained via the PANTHER Overrepresentation test (PANTHER V 16.0) against salivary proteins identified in healthy subjects (deposit in SalivaTecDB [10,15,16]). Only the biological processes significantly altered with *p* < 0.05 were considered enriched. The binomial test with Bonferroni correction was used.

A bioinformatic approach, the PLS-DA model, was used to select the proteins capable of classifying the 2 groups of samples, based on variable influence on projections (VIP) values and FDR indexes. Proteins with a VIP value higher than one, an FDR of 5% and a *p*-value < 0.05 were selected for the functional analysis. These proteins functions were analyzed by a biological processes’ enrichment analysis using the FunRich tool. Our dataset was analyzed against the FunRich curated database background. The gene ontology annotations including biological processes, the Gene Ontology database, HPRD, Entrez Gene and UniProt, were used to build the FunRich background database [15].

### 2.8. Human–SARS-CoV-2 In Silico Interactomics Analysis 

The protein–protein interactions (PPIs) derived between humans and SARS-CoV-2 proteins were predicted using the OralInt tool developed by our group [8,10], which allows the prediction of interspecies protein interactions. The input data was the human saliva proteome described in previous sections (642 proteins) as of July 2022 and the SARS-CoV-2 reference proteome deposited in UniProt [17] (14 proteins).

The predicted interactions were categorized and assessed based on the prediction score (0.9–1.0: very high confidence; 0.7–0.9: high confidence; 0.4–0.7: medium confidence; 0.1–0.4: low confidence). Interactions with scores of less than 0.7 were eliminated (Appendix A). 

A network of predicted high confidence PPIs (score ≥ 0.7) was created Using Cytoscape v.3.9.1 [18]. Network analysis was performed using Cytoscape’s Network Analyzer Tool to ease data interpretation, and quantitation data obtained by mass spectrometry were used to represent the degree of alteration of proteins after infection with SARS-CoV-2. 

A network functional analysis was also performed using the Reactome Pathway enrichment (updated 8 February 2022) with ClueGo v2.5.8 + CluePedia v.1.5.9 [19] to catalogue the disrupted system processes in the human oral environment caused by the SARS-CoV-2 infection.

### 2.9. Statistical Analysis

A Kruskal–Wallis test was performed to select the proteins statistically different between the COVID+ and COVID- samples. Dunn’s test of multiple comparisons, with Benjamini–Hochberg *p*-value adjustment, was performed to determine in which comparisons statistical differences were observed. The Mann–Whitney test was used for the binary comparisons. The multivariate analysis was performed in MetaboAnalyst as previously described [6].

## 3. Results and Discussion

We hypothesized that SARS-CoV-2 induces characteristic proteome changes that can be detected in the saliva of infected patients. We propose a methodological approach for the functional analysis of altered salivary proteins in COVID-19 identified by mass spectrometry to identify or clarify molecular changes associated with COVID-19. These molecular changes may shed light on viral particle entry mechanism, salivary markers for new diagnostic approaches (protein quantification), or the association of infection to host response. 

The proteomic analysis of the 10 saliva samples (COVID+ and COVID-) resulted in the identification of 925 proteins with a 5% FDR (Appendix A), from which 642 have a *p* < 0.05 (Appendix A). 

### 3.1. Protein Functional Analysis

Considering the initial dataset of 925 proteins, a pool of 642 were selected with 5% FDR to perform an enrichment analysis (PANTHER Overrepresentation test, Figure 1). The enrichment analysis resulted in 39 biological processes significantly altered with a 2.5-fold enrichment. Interestingly, three biological processes had over a four-fold enrichment, indicating that these are over-represented in our dataset: Immunoglobulin Production, Production of Molecular Mediator of Immune Response and Cell Recognition.

Twenty-six proteins contribute significantly to the PLS-DA model with a PC1 VIP score > 1.0 (Table 1): of these, twenty-two proteins are more abundant and four are less abundant in COVID+ samples than in healthy individual samples. Inter-alpha-trypsin inhibitor heavy chain H1 and Annexin A7 stand out by being the least and the most abundant in this dataset when comparing the two sample groups.

Biological processes enrichment analysis (FunRich tool) of the PLS-DA relevant proteins showed that these proteins are involved in five main biological processes: Apoptosis, Energy Pathways, Immune Response, Protein Metabolism and Transport, in which 12 proteins are involved (Figure 1 and Figure 2, Table 1). Transport is the process with the most proteins mapped (four), and apoptosis is the process with the least (one).

#### 3.1.1. Apoptosis

A key component of a host’s defense against viral infections is cell death [19]. Apoptosis, pyroptosis and necroptosis are the three primary types of cell death that occur in virus-infected cells, and each is regulated by a different collection of host proteins [20].

Our results suggests that the p53 apoptosis effector, a component of intercellular desmosome junctions, was the main contributor for the observed dysregulation of apoptosis (Figure 2), playing a role as an effector in the TP53-dependent apoptotic pathway [17]. Apoptosis is involved in COVID-19: the open reading frame (ORF) 3a protein of SARS-CoV (SARS 3a) can induce caspase activation [19]. Simultaneously, it is known that p53 plays a role in SARS-CoV-2 ORF3a-induced apoptosis [21]. So, in COVID-19, p53 interacts with the viral particles of SARS-CoV-2 at two different levels: a direct regulatory activity of p53 on SARS-CoV-2 replication, shared by the other coronaviruses, and more notably, the ability of SASR-CoV-2 to up-regulate the major p53 inhibitor MDM2. This down-regulation of the basal levels of p53, described as part of the survival strategy of SARS-CoV-2, is shared by other SARS-CoV viruses [22], and leads to perturbation of the tissue homeostasis, as mentioned above. The physiological significance of p53 basal activity is still emerging, as p53 suppression is linked to the disruption of tissue hemostasis. Low basal levels of p53 have been associated with respiratory disorders, suggesting a protective role for p53 in vascular hemostasis and inflammation of the lungs [23].

Our results show that in saliva, p53 apoptosis effector is less abundant (FC −2.52) in patients with COVID-19, corroborating the results obtained by Cui et al., 2021, and suggested by Milani et al., 2022 [22,23,24].

#### 3.1.2. Energy Pathways

The risk of death by SARS-CoV-2 infection is higher in patients with energy metabolism-related chronic disorders, including diabetes [25]. Our analysis showed that there is a dysregulation of energy processes after SARS-CoV-2 infection, which may help to explain the increased mortality in these patients [25,26]. In fact, according to preliminary clinical data on COVID-19 patients, individuals with type 2 diabetes and other metabolic disorders that impair general metabolic health are more likely to experience a more severe infection course than those who were metabolically healthy before contracting the virus [26]. This could be due to the impact of SARS-CoV-2 infection and disease outcome based on the energy metabolism equilibrium [25].

The oligosaccharyl transferase 48 kDa (OST) subunit and inositol monophosphatase 1 are the main contributors for the dysregulation of energy pathways processes observed in our analysis. Oligosaccharyl transferase 48 kDa subunit catalyzes the initial transfer of a glycan from the lipid carrier dolichol-pyrophosphate to an asparagine residue, the first step in protein N-glycosylation [17]. Oligosaccharyl transferase gene DDOST is related with the innate immune system and with SARS-CoV-2 infection, and was identified interacting with nonstructural proteins (NSP4) important for virus replication [27]. In fact, reactome analysis (Figure 3) shows a maturation of the SARS-CoV-2 spike protein by N-glycosylation that seems to involve OST.

It has been hypothesized that the blockage of the active site of oligosaccharyl transferase can significantly inhibit the infection of both SARS-CoV-2 and its variants [29,30,31]. The increase on the quantity of oligosaccharyl transferase (FC 3.15) in saliva samples of COVID+ patients agree with data suggesting the involvement of OST in spike protein glycosylation. To our knowledge, our study is the first to identify and quantify this protein in the saliva of patients with COVID-19.

Inositol monophosphatase 1 (IMPA2) is responsible for the provision of inositol required for synthesis of phosphatidylinositol and polyphosphoinositides, being an important modulator of intracellular signal transduction via the production of the second messengers myoinositol 1,4,5-trisphosphate and diacylglycerol [17]. Until now, there is no known direct relation with COVID-19, although it has been related with pulmonary complications with an important role in pulmonary arterial hypertension [32]. Nonetheless, the increase (FC 1.64) of inositol monophosphatase 1 on the saliva of COVID-19 patients might be related to a defense response to the respiratory infection. It is known that myoinositol promotes the maturation of pulmonary surfactant phospholipids, regulating the synthesis of type II pneumocytes [33]. Inositol promotes a mechanical stabilization of cell shape enabling alveolar cells to counteract collapsing forces.

#### 3.1.3. Immune Response

An efficient elimination of invading pathogens requires a good synergy between innate and adaptive immune responses [34]. A highly ordered cellular and molecular cascade is involved in the immune response controlling the balance of viral eradication vs. immunological harm [35]. In fact, SARS-CoV-2 infection-associated immune responses are central to the pathogenesis of COVID-19 [36].

Our functional analysis shows that galectin-3 binding protein and CD59 glycoprotein may be key modulators of the immune response biological process in COVID-19. Galectin-3 binding protein promotes integrin-mediated cell adhesion and may stimulate host defense against viruses and tumor cells [17]. Galectin-3 proteins were proposed to act as alarmins due to amplifying inflammatory responses during sepsis and several types of infection [37,38,39]. Interestingly, a high abundance (higher than 30.99 ng/mL) determined in serum can predict severity disease state [40]. Furthermore, galectin-3 orchestrates the inflammation response by activating innate immune cells and releasing different cytokines, including IL-6 and TNF alpha which are present in high concentrations in severe COVID-19 cases [36,37,38]. Galectin-3 has also been proposed as a biomarker of the inflammatory status in COVID-19 patients. Cervantes-Alvarez et al., 2022 [39] have suggested that plasma levels of this protein have shown a strong correlation with lung fibrosis progression, a consequence of COVID-19. Additionally, galectin-3 has been found elevated in the serum of COVID-19 patients [41]. In our study, galectin-3 was more abundant in COVID-19 group samples, with a fold change of 4.80.

CD59 glycoprotein is a potent inhibitor of the complement membrane attack complex (MAC) action. CD59 acts by binding to the C8 and/or C9 complements of the assembling MAC, thereby preventing incorporation of the multiple copies of C9 required for complete formation of the osmolytic pore. The CD59 glycoprotein is involved in signal transduction for T-cell activation complexed to a protein tyrosine kinase [17]. The activity of CD59 in COVID-19 stills unveiled. However, many enveloped viruses are assembled in lipid rafts, which contain CD59 proteins [42]. The lipid rafts are a cholesterol-rich domain found on the surface of cells, normally aggregating the TCR-antigen ligation site. This structure ensures the optimal immunological synapse on T-cell activation and antigen-specific signaling [43]. It is also known that CD59 inhibits the formation of complement membrane attack complex pores in the membranes of expressing cells: it is a ‘suicide inhibitor’ [44]. Our results shows that CD59 was more abundant in the COVID-19 sample group (FC 1.84). This result is not in agreement with Ramlal et al., 2020, who claimed that there is an under regulation of the CD59 gene during the COVID-19 infection [45]. Ramlal’s investigation cannot be directly compared with our proteomic profiling of saliva, since Ramlal characterized the transcriptome of nasopharyngeal swabs. In this study, SNPs from CD55 were found to negatively affect expression levels which are associated with a high risk of negative outcome [46].

#### 3.1.4. Protein Metabolism

During infection, alterations in host metabolism occur at all levels—cellular, tissue, organ and physiological. Recent data from different infectious illnesses have shown that metabolic processes are significant mediators of host defense systems that guard against the physiological damage that happens during infections and subsequently permit survival. Impaired protein metabolism is a metabolic abnormality linked with severe COVID-19, characterized as increased protein and muscle breakdown, reduced muscle synthesis and increased synthesis of acute phase proteins [47].

The enrichment analysis showed that kallikrein 11 protein, inter-alpha-trypsin and alpha-2-macroglobulin are the main contributors for the dysregulation of this biological process.

Kallikrein 11 is a multifunctional protease that cleaves the kallikrein substrate and trypsin [17]. Bradykinin production by the kallikrein/kinin system is involved in acute respiratory distress syndrome (ARDS) of bacterial sepsis origin and may be a driver of the COVID-19 ARDS-like lung injury. These peptides have chemotactic properties, induce increased vascular permeability, and activate immune and endothelial cells leading to inflammation and cyto-/chemokine expression [48].

Kallikrein 11 (KLK11) has not been associated with COVID-19 until now, but there are a few studies showing the relation of the kallikrein–renin pathway with the infection by SAR-CoV-2. Correlations of this system, specifically the consumption of pekilocerin, with patient death due to respiratory failure have already been reported [49].

Lipcsey et al., 2021, suggested that the strong activation of the kallikrein/kinin system can be the main driver of the ARDS-like condition in COVID-19 [49]. This pathway was also linked to COVID-19 by Carvalho et al. (2021), relating the activation of a kallikrein-like effect (TMPRSS2 transmembrane serine protease 2) to an increase in the production of bradykinin, leading to inhibition of the ACE2 pathway, the main entry of SARS-CoV-2 in the cells [48,50]. This interaction was described also by Sidarta-Oliveira et al., 2020, defining its expression mainly to alveolar area [49]. The kallikrein-11 increase (FC 2.61) in saliva of COVID-19 patients may be related to an active virus replication and cell entry.

Inter-α-trypsin inhibitor heavy chain H1(ITIH1) acts as a carrier of hyaluronan in serum or as a binding protein between hyaluronan and other matrix proteins, including those on cell surfaces in tissues to regulate the localization, synthesis and degradation of hyaluronan, which are essential to cells undergoing biological functions. ITIH1 contains a peptide that potentially stimulates a broad spectrum of phagocytotic cells [17]. The function of inter-α-trypsin inhibitor heavy chain 1 has not yet been described in COVID-19. Nonetheless, in severe COVID-19 patients, ITIH1 is downregulated [50] and is more abundant in COVID-19 survivors group compared with the and non-survivors [51]. Interestingly, according to our work, inter-α-trypsin inhibitor heavy chain H1 was found to be less abundant in the saliva of COVID-19 patients than in healthy patients (FC −8.28), which could be related to the severity of the disease.

Alpha-2-macroglobulin (α2-M) inhibits all four classes of proteinases by a unique ‘trapping’ mechanism. This protein has a peptide stretch, called the ‘bait region’, which has specific cleavage sites for different proteinases. When a proteinase cleaves the bait region, a conformational change is induced in the protein trapping the proteinase. The entrapped enzyme remains active against low molecular weight substrates (activity against high molecular weight substrates is greatly reduced). Following cleavage in the bait region, a thioester bond is hydrolyzed and mediates the covalent binding of the protein to the proteinase [17]. An interaction with α2-M was tested for human immunodeficiency virus type 1, and a cleavage was shown in the bait region [52,53]. Alpha-2-macroglobulin is known by the capacity of maintaining the hemostatic balance and moderating innate immunity [54,55], preventing structural damage during inflammation by inhibiting the expression of proteases by leukocytes that have been activated and also suppressing proteases released by invasive microbes [56]. This protein also participates in the regulation of inflammatory mediators. In COVID-19, α2-M is attached on the luminal surface of endothelial cells [57]. Seitz et al., 2020, suggested that a higher abundance of α2-M during SARS-CoV-2 infection in childhood favors a positive disease outcome [55]. Oguntuyo et al. [58] determined that a higher abundance of alpha-1-antitrypsin and α2-M in serum could inhibit the SARS-CoV-2 particle invasion. The observed downregulation of alpha-2-macroglobulin in COVID-19 saliva (FC –2.91) agrees with an active infection.

#### 3.1.5. Transport

To complete their life cycles, beta-CoVs as SARS-CoV-2 and HCoV-OC43, use an overlapping collection of host factors. These are genes involved in transport and biosynthetic processes [59]. These findings imply that SARS-CoV-2 is dependent on unique intracellular host factors and complexes that regulate intracellular transport [59].

Rab GTPases and Rab GTPase regulatory proteins, which govern intracellular transport, anchoring and exocytosis of secretory vesicles, are another group of important SARS-CoV-2 host factors [59].

Our enrichment analysis showed that annexin A7, proteolipid protein 2, transcobalamin 1 and lactotransferrin are the main contributors for the dysregulation of transport (Table 1). 

Annexin A7 (ANXA7) is a calcium/phospholipid-binding protein which promotes membrane fusion and is involved in exocytosis [17]. Annexin A7 is part of a calcium-binding protein family implicated in vesicle transport and apoptosis. Exocytosis, glutamate release and N-Methyl-D-aspartic acid (NMDA) trafficking are all functions of ANXA7 [60]. This supports the theory that there can be molecular mimicry of SARS-CoV-2 targeting ANXA7-expressing cells in the brain, resulting in neurological injury [60]. The interaction of human annexin A7 and SARS-CoV-2 ORF1ab was described by Venkatakrishnan et al., 2020 [61], which is in accordance with the higher levels of annexin A7 (FC 6.40) in COVID-19 patients. Annexins A2 and A5 were associated with COVID-19 [62]. Annexin A2 is critical for fibrinolysis in the lung by acting as a co-receptor that activates endogenous tissue plasminogen activator (t-PA) to lyse clots and promote fibrin clearance. Its inhibition might also explain the diffuse alveolar damage, ARDS and pulmonary fibrosis seen in severe cases of COVID-19 [62]. 

The proteolipid protein 2 (PLP2) [17,63] was defined as important for viral progression and linked to regulation of mitochondrial function in SARS-CoV-2 [64]. Although the link between PLP2 and COVID-19 is not well established yet, we found that this protein is more concentrated in COVID-19 saliva samples (FC 3.05). This can represent a suppression of antiviral mechanisms and active infection progression.

Transcobalamin 1, also known as haptocorrin, is encoded by the *TCN1* gene. This gene encodes a member of the vitamin B12-binding protein family [17]. The association of TCN1 with COVID-19 was defined by its involvement in neutrophil-mediated immune responses [65,66]. Distinct types of enzymes related to limiting host infection are expressed by neutrophil granulocytes. The cytotoxic chemicals influence the inflammatory response when they are released from the granules. As commonly related, COVID-19 patients manifest perivascular infiltrates surrounding the capillaries in the lungs. In these patients, TCN1 is up-regulated [66], as it is in COVID19+ saliva (FC 2.01). The correlation of TCN1 with type 2 diabetes mellitus on COVID-19 outcome is being studied [26]. Higher levels of vitamin B12 were related with expanded seriousness of COVID-19 [67]. On the other hand, in a plasma proteome profiling of COVID-19 contaminated patients, transcobalamin 2 was found to be elevated compared to the control group [68].

Lactotransferrin is known by its broad-spectrum antiviral activity as well as immunomodulatory and anti-inflammatory actions [69]. It can simultaneously counteract the inflammatory and iron homeostasis disorders caused by bacterial and viral attacks [70]. Lactotransferrin has been associated to COVID-19 pathology as a possible treatment [69]. Recent in vitro studies demonstrated its antiviral activity against SARS-CoV-2 [69,70,71]. The lower quantity of lactotransferrin in the saliva of COVID-19+ patients (FC –1.53) is related to the presence of an active viral infection. 

### 3.2. Other Proteins

PLS-DA analysis is a well-known tool for analyzing multidimensional data. It incorporates a discriminatory algorithm that identifies the variables that allow the distinction between groups. A VIP score measures a variable’s importance in the PLS-DA model, summarizing the contribution of each variable to the model. Proteins with a VIP > 1, but that do not account for the enrichment of the biological processes, are listed in Table 1, and include several immunoglobulins in patients with COVID-19. Until now, none of the identified immunoglobulins identified were related to COVID-19. Interestingly a similar immunoglobulin, the lambda variable 3–25, was identified as a COVID-19 characteristic protein that may indicate the progression of the two stages of the COVID-19 disease [72].

Ras-related C3 botulinum toxin substrate 1 (RAC1) is a plasma membrane-associated small GTPase that, when active, binds to a variety of effector proteins regulating cellular responses from secretory processes to phagocytosis of apoptotic cells, and neurons adhesion, migration and differentiation [17]. Ras-related C3 botulinum toxin substrate 2 (RAC2) is involved in inflammation-mediated lung damage [73]. The link to COVID-19 was assigned by Wang et al., 2020 [74], with neutrophil activation after the SARS-CoV-2 invasion by the RAC2 gene.

Thymidine phosphorylase (TYMP) encodes an angiogenic factor which promotes angiogenesis in vivo and stimulates the in vitro growth of a variety of endothelial cells [17]. TYMP has a role in the systemic immune response to SARS-CoV-2 infection. Because red blood cells do not produce TYMP and TYMP is not secreted, it is hypothesized that that the elevated plasma TYMP concentration in COVID-19 patients was caused by either platelet-rich thrombolysis or organ injury. TYMP expression is high in both platelets and the lung [75], as well as in saliva (5.89 times more abundant than in the saliva of healthy patients, Table 1).

Ly6/PLAUR domain-containing protein 2 (LYPD2) is predicted to be extracellular or in the plasma membrane [17]. LYPD2 is highly expressed in transcriptional profiles in non-classical monocytes from patients with COVID-19 [76]. The gene encoding for this protein is differentially expressed in two neutrophil subpopulations [77]. Non-classical monocytes may ingest virally infected apoptotic endothelial cells [78]. The 2.62-fold increase of LYPD2 in COVID-19 saliva samples may represent an active immune system fighting against infection in the COVID+ sample group, but further validation is needed.

Transmembrane protein 165 (TMEM165) is a transmembrane protein produced in fibroblasts that has been linked to bacterial infections in the lungs [17]. SARS-CoV-2 interactome reveled the interaction between ORF3a and TMEM165, which is consistent with the knowledge that ORF3a hijacks the HOPS (homotypic fusion and vacuole protein sorting) complex and RAB7, required for membrane contact between autophagosomes and lysosomes for autolysosomes to develop [79]. The increase on TMEM165 abundance in the saliva of COVID-19 patients (FC 2.42) is consistent with an active infection.

Metabolism regulating signaling molecule B (FAM3B) induces apoptosis of alpha and beta cells in a dose- and time-dependent manner [17]. FAM3B expression in the endocrine pancreas is stimulated by hyperglycemia and proinflammatory cytokines. FAM3B promotes insulin secretion under physiological settings, but it is also a secreted cytokine-like protein that can cause cellular death. Increased FAM3B production is linked to pancreatic cell dysfunction, hyperglycemia and insulin resistance, implying a role in the control of glucose and lipid metabolism. Fraser et al. [80] has suggested FAM3B as a protein biomarker able to predict survival/death of patients with COVID-19. In this study, COVID-19 patients had a 2.25-fold increase of FAM3B.

Ras-related protein Rab-6A is a protein transport and regulator of membrane traffic from the Golgi apparatus towards the endoplasmic reticulum [17], found interacting with SARS-CoV-2 RNA in infected human cells [81]. This interaction was also mentioned in a study from Pereira et al., 2021 [82]: RAB6A was found interacting with SARS-CoV-2 cells associated with vesicle trafficking [81]. Viral infection may alter the host cell’s exosomal-loading processes, resulting in alterations in the protein and nucleic acid content of extracellular vesicles (EVs). It indicates that infected cells produce modified EVs rather than relying on viral content. As a result, as compared to EVs that do not include infected cells, these changed EVs may influence the host immunological response [83]. Our results show RAB6A had a 2.18-fold change, indicating a vesicle trafficking compromise.

V-Set and immunoglobulin domain containing 10-like (VSIG10) is expressed in the esophagus (and many other oral and digestive tissues) and is involved in anti-inflammatory responses. It is the first time that this protein is related to COVID-19 (two-fold increase in COVID-19 patients).

Calcium-binding protein 39 (CAB39) enables kinase binding activity and protein serine/threonine kinase activator activity [17]. This protein was associated with SARS-CoV by a functional enrichment pathway [84] and is less abundant in SARS-CoV-2 blood samples [85]. We detected a 1.77-fold increase of CAB39, potentially indicating that different body fluids may show different trends for the same protein.

Prohibitin 2 (PHB2) is involved in defense response to viruses [17]. It may contribute to viral replication by arresting normal host cell functions [86]. The interaction of nnon-structural protein 2 (NSP2) of SARS-CoV with prohibitin 2 was shown by Cornillez-Ty CT. et al. (2009) [87]. It is hypothesized that coronavirus proteins may impact key mitochondrial functions such as respiration, but also lipid homeostasis and innate immunity. Indeed, mitochondria are involved in both lipogenesis and lipolysis, and prohibitin expression has been shown to impact lipid accumulation and degradation [88]. 

### 3.3. Human–SARS-CoV-2 In Silico Interactomics Analysis

The human–SARS-CoV-2 interactome was predicted using OralInt v 2.0 [12]. Protein quantitation data obtained by mass spectrometry were used to represent the degree of alteration of proteins after infection with SARS-CoV-2.

OralInt [12] (https://bioinformatics.ua.pt/software/oralint/, accessed on 22 February 2022) is an algorithm designed for interactome prediction: it allows the identification of key proteins involved in pathologies which may be used in diagnostics or as therapeutic targets, as well as the proposal of microbial infection mechanisms. 

The in silico interactomics analysis predicted a total of 10,914 protein–protein interactions (PPIs), with 263 of these PPIs showing a high level of confidence (SCORE ≥ 0.7). The 14 virus proteins interact with 100 human proteins found altered in the saliva of COVID-19 patients (Figure 4). Only 5 of these 100 human proteins were identified in SARS-CoV-2 infection pathway and related pathways by the PathCards (genecards.org accessed on 22 February 2022): Ubiquitin-60S ribosomal protein L40, beta-2-microglobulin, ubiquitin-conjugating enzyme E2 variant 1, 14-3-3 protein beta/alpha and dolichyl-diphosphooligosaccharide protein glycosyltransferase subunit 1. The remaining 95 proteins might be related to not yet known molecular changes of SARS-CoV-2 infection.

#### 3.3.1. Viral Hub Proteins

Viral hub proteins interact mainly with proteins from human immune system, suggesting a direct influence on inflammation and host defense modulation. It is worth noting that this is the first time that ADP-ribosylation factor 4 (cellular trafficking) has been associated with COVID-19.

The hub proteins ORF1a and rep are involved in a high number of interactions with human proteins (histone H4 (H4); inter-alpha-trypsin inhibitor heavy chain H1; lumican (LUM); myeloperoxidase (MPO); beta-2-microglobulin (B2M); ADP-ribosylation factor 4 (ARF4) and thioredoxin domain-containing protein 17 (TXNDC17); Figure 4). ORF1a and rep are multifunctional virus proteins involved in the transcription and replication of viral RNAs. Both proteins inhibit antiviral response triggered by innate immunity or interferons [88,89,90] and may play a role in the modulation of the host cell survival signaling pathway. 

ORF1a and rep interact with histone H4 (H4). Histone H4 has the most potent antiviral activity among histones and has been shown to promote neutrophil activation and consequently the early phase of the innate immune response to influenza A virus [91,92,93,94]. This study shows a reduction in histone H4 in patients with COVID-19 (FC −14.47), which may prevent an effective elimination of the virus at an early stage of infection.

Inter-alpha-trypsin inhibitor heavy chain H1 (ITIH1) (FC −8.28) contains a phagocytosis uptake signal motif which could stimulate a broad spectrum of phagocytotic cells. We hypothesize that a reduction in ITIH1 may compromise the phagocytosis of viral particles. This is supported by the reduction in apoptosis observed in the functional analysis (Figure 2) which is known to precede the phagocytosis of viruses already observed regarding influenza A virus [92]. 

Lumican (LUM) (FC −5.90) has been found to be critical for the host immune innate response [95]; therefore, lower lumican levels may act as an additional biomarker of inflammation in the COVID-19 patients. 

Myeloperoxidase (MPO) (FC −4.44) plays a role in the host defense system of polymorphonuclear leukocytes. It is responsible for microbicidal activity against a wide range of organisms (Figure 4—Innate immune system). In the stimulated PMN, MPO catalyzes the production of hypochlorous acid, in physiologic situations, and other toxic intermediates that greatly enhance PMN microbicidal activity [96]. The production of MPO can be activated by histone H4 and, since it is largely reduced in patients with COVID-19 (FC −14.47), this can lead to a reduction of MPO and, consequently, diminish the antiviral responsiveness to SARS-CoV-2.

Beta-2-microglobulin (B2M) is a component of the class I major histocompatibility complex (MHC) involved in the presentation of peptide antigens to the immune system (Figure 4). B2M is also essential for the correct subcellular distribution of both hereditary hemochromatosis protein (HFE) and hepcidin, two proteins critical for iron homeostasis [97]. An increase in B2M and consequently in hepcidin can lead to the marked hypoferremia that seems to characterize severe COVID-19, and may contribute to worsening prognosis by impairing not only response to hypoxia, but also immune function [98]. Finally, it should be considered that low iron status can theoretically impair the efficacy of COVID-19 vaccination [99]. In this study, B2M was found to be increased in the saliva of patients with COVID-19 (FC 3.78), which agrees with the work of Conca and colleagues [100], who related the serum levels of this protein with the severity of the disease. Our work demonstrates that it is possible to measure the levels of this protein in a less invasive way in saliva samples, maintaining the trend observed in serum [101].

Thioredoxin domain-containing protein 17 (TXNDC17) (FC 7.59) has not yet been linked to COVID-19, although it is known to modulate TNF-alpha signaling and NF-kappa-B activation [102], which have a role in cytokine storm syndrome, associated with greater severity in COVID-19-related symptoms. Therefore, therapeutics that reduce the levels of TXNDC17 may be considered in the treatment of COVID-19.

ADP-ribosylation factor 4 (ARF4) is a GTP-binding protein involved in protein trafficking. To our knowledge, the direct correlation between ARF4 and SARS-CoV-2 infections has not yet been established. However, there is evidence that hepatitis C virus (HCV) infection is associated with an upregulation of ARF4, which promotes HCV replication [99]. The pronounced increase in ARF4 (FC 7.37) observed in our study suggests that SARS-CoV-2 may employ a replication-promoting mechanism similar to HCV, as both are Group IV viruses ((+)ssRNA). This may provide new therapeutic targets for SARS-CoV-2 antiviral therapy.

#### 3.3.2. Human Hub Proteins

The functional analysis of the high-confidence interaction network between human saliva proteins and SARS-CoV-2 proteins shows that many of these interactions are associated with proteins from the human immune response (transcobalamin 1 (TCN1) and nucleobindin-2 (NUCB2)), mainly through the degranulation of neutrophils. Furthermore, the virus alters the host’s energy metabolism and interferes with apoptosis mechanisms (Figure 5 and Section 3.1).

The association of the NUCB2 protein (FC −3.04) with COVID-19 is not yet established. It is known that this protein may be involved in the regulation of inflammation, immune functions, host defense and apoptosis through mediating TNF-alpha receptor (TNFR1) release [103]. The decrease in NUCB2 in patients infected with SARS-CoV-2 (FC −3.04) may contribute to the cytokine storm characteristic of the most severe forms of COVID-19, as a decrease in the amount of TNFR1 can cause an increase in the production of TNF-alpha to compensate for the decrease in receptor availability. 

TCN1 (FC 2.08) is a major constituent of secondary granules in neutrophils, is involved in neutrophil-mediated immune responses (Figure 5) and has been previously found upregulated in nasopharyngeal swabs from SARS-CoV-2 patients [67]. This study shows that these alterations can also be observed in saliva samples and corroborate other studies that hypothesized that the increase in TCN1 may be related to a generalized hyperinflammatory state characteristic of COVID-19 [66].

## 4. Conclusions

Although we consider that, in the future, further validation—with larger sample groups, that allow correlating molecular data with patient symptoms—should be considered, this study is a proof of concept of the molecular aspects of SARS-CoV-2 infection applying a proteomic approach on saliva.

We have identified 26 proteins altered in patients with COVID-19, which may be indicative of alterations resulting from the disease. Fourteen had already been described in other types of samples but not in saliva. On the other hand, 10 had never been associated with SARS-CoV-2 infection before, representing new evidence of the impact of COVID-19 on the host. As expected, many of the proteins altered upon infection with SARS-CoV-2 are related to the immune system, and the interactome analysis shows that the virus directly interacts with human immune system proteins. We showed that the virus alters the host’s energy metabolism and interferes with apoptosis mechanisms. We also showed that many of the proteins that are altered upon infection with SARS-CoV-2 are involved in apoptosis, transport and signal transduction, revealing the broad spectrum of alterations that COVID-19 represents.

## Figures and Tables

**Figure 1 jcm-11-05571-f001:**
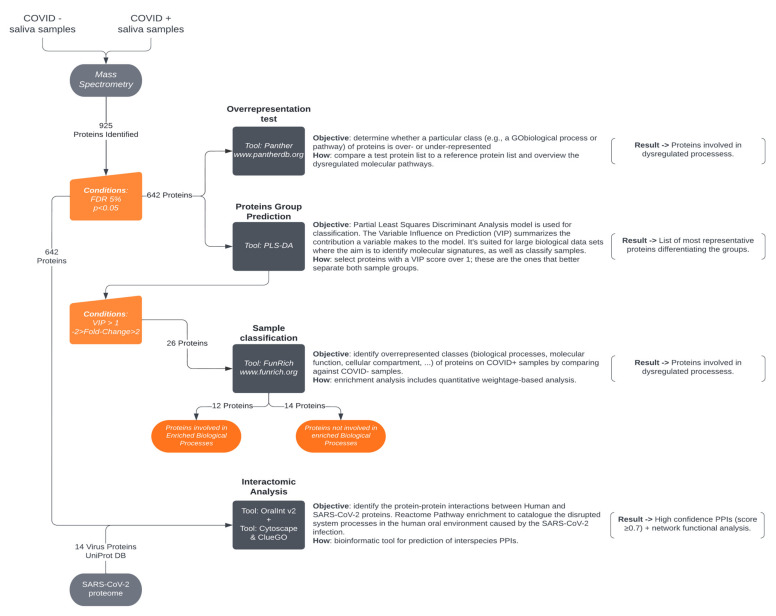
Scheme of the protein functional analysis and interactomics workflow. Proteins were identified by LC-ESI-TOF mass spectrometry. An initial dataset of 925 proteins was narrowed down via filtering by an FDR 5% and *p* < 0.05 conditions, resulting in 642 salivary proteins then analyzed (GO biological process) by Panther overrepresentation analysis. Proteins (26) with a VIP > 1 (PLS-DA model) and −2 > Fold-Change > 2 defined were used in sample classification analysis with the FunRich tool. The enrichment resulted in 12 proteins with dysregulated processes and with *p* < 0.05. Interactome analysis on the narrowed down dataset against the SARS-CoV-2 proteome.

**Figure 2 jcm-11-05571-f002:**
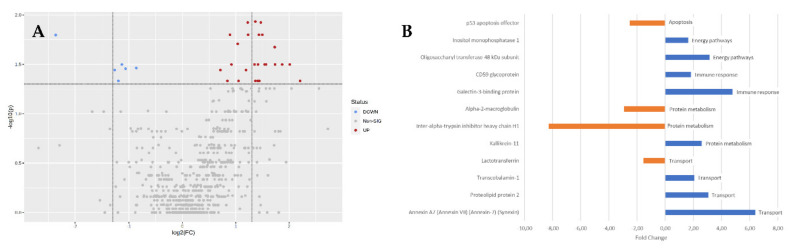
(**A**) Graphical representation of the protein distribution by protein expression (*X* axis) and VIP score (*Y* axis) variables in a Volcano plot. Blue and red dots represent up and down regulated proteins. Gray dots represent t=e proteins without regulation data. (**B**) Biological processes enrichment analysis graphical representation of the proteins with VIP score > 1 against the FunRich background database. The protein names are listed in the Y axis, with the respective fold change on the *X* axis. The biological process is indicated next to the graphic bar in A→Z order.

**Figure 3 jcm-11-05571-f003:**
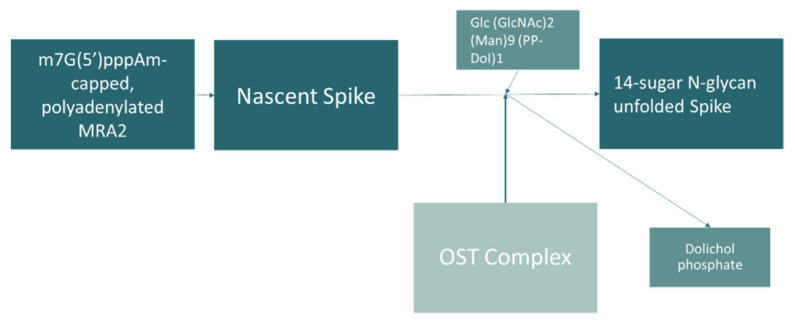
Pathway of spike protein glycosylation, according to Reactome.org [28]. DDOST gene relation with SARS-CoV-2 infection: maturation of SARS-CoV-2 spike protein by N-glycosylation involving the oligosaccharyltransferase (OST) complex. Mammalian cells express OST complexes that contain a catalytic subunit and accessory proteins as the dolichyl-diphosphooligosaccharide–protein glycosyltransferase 48 kDa subunit (DDOST). DDOST catalyzes the initial transfer of a glycan from the lipid carrier dolichol-pyrophosphate to the nascent polypeptide chain. Image adapted from reactome.org (accessed on 22 February 2022).

**Figure 4 jcm-11-05571-f004:**
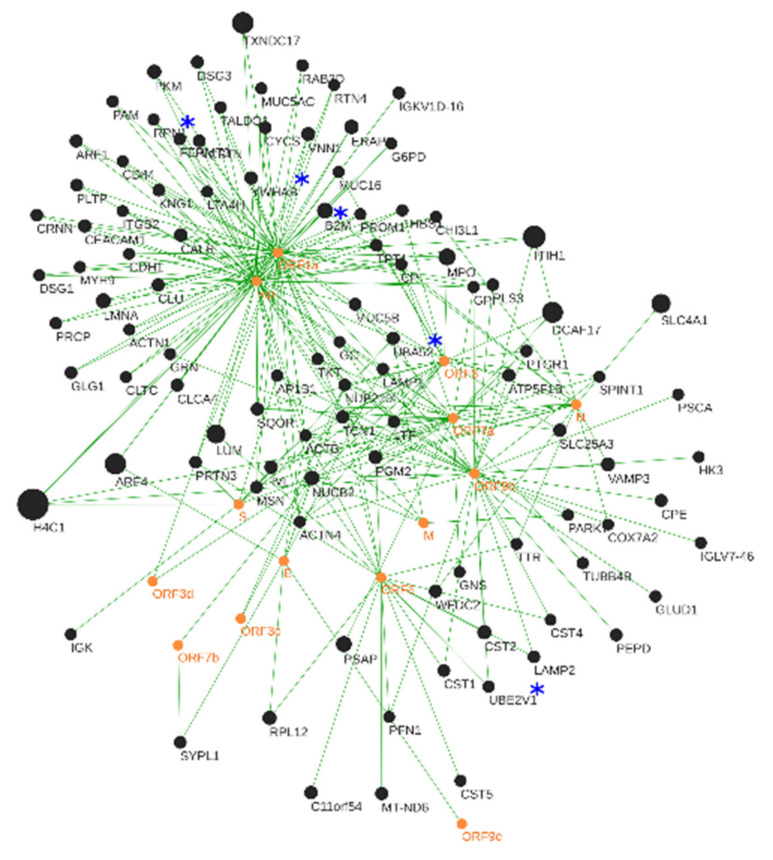
High-confidence network of interactions (263 interactions) resulted from the input of 642 human proteins and the SARS-CoV-2 reference proteome (14 proteins) predicted by OralInt. Network visualization was done using Cytoscape. SARS-CoV-2 proteins in orange, human proteins in black. Hub proteins in bold. Node size is proportional to the absolute value of the fold change. * Represent the five proteins already identified in the SARS-CoV-2 pathway by PathCards.

**Figure 5 jcm-11-05571-f005:**
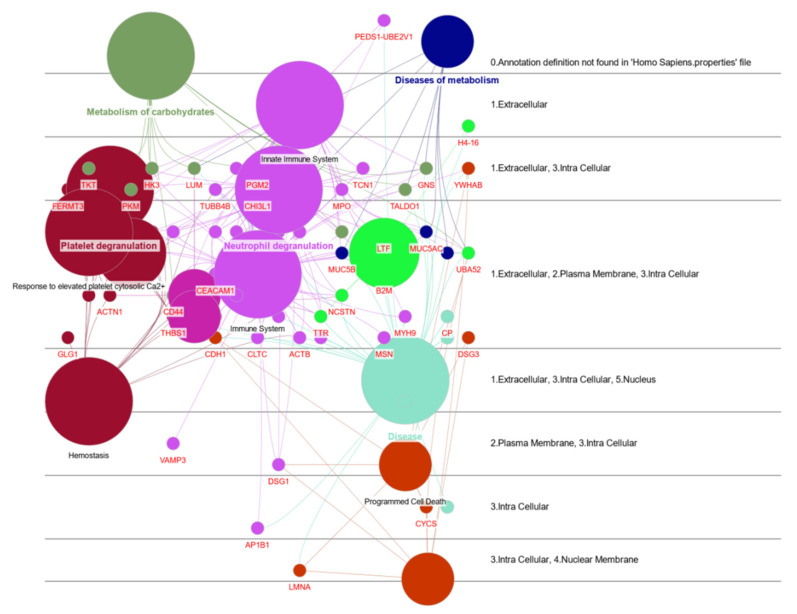
Functional analysis of the high-confidence interaction network of human saliva proteins with SARS-CoV-2 proteins. Reactome pathway analysis, using ClueGo + CluePedia; visualization with Cerebral View.

**Table 1 jcm-11-05571-t001:** List of the proteins with a VIP score > 1. These are the proteins that better differentiate COVID+ and COVID- samples. Shown are the UniprotKB AC code, protein names, gene names, fold change (COVID+/COVID-) and the biological process defined by the enrichment analysis. * Proteins involved in enriched biological processes.

UniprotKB AC	Protein Name	Gene Name	Fold Change	Biological Process
P19827	Inter-alpha-trypsin inhibitor heavy chain H1	*ITIH1*	−8.28	* Protein metabolism
P01023	Alpha-2-macroglobulin	*A2M*	−2.91	* Protein metabolism
Q96FX8	p53 apoptosis effector	*PERP*	−2.52	Apoptosis
P02788	Lactotransferrin	*LTF*	−1.53	Transport
Q99623	Prohibitin-2	*PHB2*	1.61	Mitochondrion organization
P29218	Inositol monophosphatase 1	*IMPA1*	1.64	* Energy pathways
Q9Y376	Calcium-binding protein 39	*CAB39*	1.77	Protein serine/threonine kinase activity
P13987	CD59 glycoprotein	*CD59*	1.84	* Immune response
P15153	Ras-related C3 botulinum toxin substrate 2	*RAC2*	1.88	Regulation of respiratory burst
Q86VR7	V-set and immunoglobulin domain-containing protein 10-like	*VSIG10L*	1.99	Cell adhesion molecule
P20061	Transcobalamin-1	*TCN1*	2.08	* Transport
P01706	Immunoglobulin lambda variable 2–11	*IGLV2–11*	2.17	Response to bacterium
P20340	Ras-related protein Rab-6A	*RAB6A*	2.18	Antigen receptor-mediated signaling pathway
P58499	Protein FAM3B	*FAM3B*	2.25	Antimicrobial response protein
Q9HC07	Transmembrane protein 165	*TMEM165*	2.42	Humoral immune response
Q9UBX7	Kallikrein-11	*KLK11*	2.61	* Protein metabolism
Q6UXB3	Ly6/PLAUR domain-containing protein 2	*LYPD2*	2.62	Mitotic cell cycle
Q04941	Proteolipid protein 2	*PLP2*	3.05	* Transport
P63000	Ras-related C3 botulinum toxin substrate 1	*RAC1*	3.06	Regulation of cell shape
A0A0C4DH68	Immunoglobulin kappa variable 2–24	*IGKV2–24*	3.09	Immune response
P39656	Oligosaccharyl transferase 48 kDa subunit	*DDOST*	3.15	* Energy pathways
P0DP04	Immunoglobulin heavy variable 3–43D	*IGHV3–43D*	3.37	Defense response to bacterium
Q08380	Galectin-3 binding protein	*LGALS3BP*	4.80	* Immune response
P0DOX2	Immunoglobulin alpha-2 heavy chain		4.86	Immune response
P19971	Thymidine phosphorylase	*TYMP*	5.89	Mitochondrial genome maintenance
P20073	Annexin A7	*ANXA7*	6.40	* Transport

## Data Availability

The mass spectrometry proteomics data have been deposited to the ProteomeXchange Consortium via the PRIDE [104] partner repository with the dataset identifier PXD036571.

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
