# Peer review of "COVID-19 Salivary Protein Profile: Unravelling Molecular Aspects of SARS-CoV-2 Infection"

_jcm, 2022, doi:10.3390/jcm11195571_

Round 1
Reviewer 1 Report
An exciting and well-written manuscript assessing salivary protein profiles during COVID-19. I am enclosing two minor remarks below:
1. I ask the authors to briefly (e.g., in the table) characterize patients with COVID-19.
2. I have no comments about the laboratory methods; they are adequately characterized.
3. Results - detailed and well described. Congratulations to the authors.
4. Abstract - please extend the results and shorten the background of the study.
5. Please briefly discuss the limitations of the study.
6. Please quote the following manuscripts:
doi: 10.3390/pathogens9060493.
doi: 10.3390/jcm9051491.
Reviewer 2 Report
In this manuscript, Esteves et al investigated the salivary protein profile of COVID-19 patients by combing a hybrid MS-based proteomics and in silico interactomics strategy. These results provided the general analysis of the differential proteome and interspecies PPIs between humans and SARS-CoV-2. Five dysregulated biological processes were identified in COVID-19 proteome profile. The authors talked about each of these differentiated proteins in these dysregulated biological processes equally by summarizing previous studies, which makes this manuscript much like a review paper and lack of highlights. Similarly, I don’t think each of these PPIs is deserved a solo paragraph. Emphasis on the previous reported PPIs and analysis on the newly emerged PPIs might provide a clearer take-home messenger for the readers.
In addition, some minor issues are needed to be fixed.
1. No citation of Figure 5;
2. Figure 3: Spyke or Spike?
3. Unify the name of SARS-CoV-2.
Round 2
Reviewer 2 Report
The authors addressed my major concern by modifing the results/discussion section and highlighting the main findings and I recommend publication.